# Enhancing Graph Invariant Learning from a Negative Inference Perspective

**Kuo Yang** [1 2 †]  **Zhengyang Zhou** [1 2 †]  **Qihe Huang** [1 2]  **Wenjie Du** [1 2 *]  **Limin Li** [1 2]  **Wu Jiang** [3]  **Yang Wang** [1 2 *]

## Abstract

The out-of-distribution (OOD) generalization challenge is a longstanding problem in graph learning. Through studying the fundamental cause of data distribution shift, i.e., the changes of environments, significant progress has been achieved in addressing this issue. However, we observe that existing works still fail to effectively address complex environment shifts. Existing practices place excessive attention on extracting causal subgraphs, inevitably treating spurious subgraphs as environment variables. While spurious subgraphs are controlled by environments, the space of environment changes encompass more than the scale of spurious subgraphs. Therefore, existing efforts have a limited inference space for environments, leading to failure under severe environment changes. To tackle this issue, we propose a negative inference graph OOD framework (NeGo) to broaden the inference space for environment factors. Inspired by the successful practice of prompt learning in capturing underlying semantics and causal associations in large language models, we design a negative prompt environment inference to extract underlying environment information. We further introduce the environment-enhanced invariant subgraph learning to effectively exploit inferred environment embedding, ensuring the robust extraction of causal subgraph in the environment shifts. Lastly, we conduct a comprehensive evaluation of NeGo on real-world datasets and synthetic datasets across domains. NeGo outperforms baselines on nearly all datasets, which verify the effectiveness of our framework.

†Equal Contribution. [1]University of Science and Technology of China (USTC), Hefei, China [2]Suzhou Institute for Advanced Research, USTC, Suzhou, China [3]China Mobile Communications Group Co.,Ltd. Correspondence to: Yang Wang* <angyan@ustc.edu.cn>, Wenjie Du* <duwenjie@ustc.edu.cn>.

*Proceedings of the 42nd International Conference on Machine Learning*, Vancouver, Canada. PMLR 267, 2025. Copyright 2025 by the author(s).

## 1. Introduction

Graph Neural Networks (GNNs) have emerged as the predominant approach for encoding graph data (Kipf & Welling, 2016; Xu et al., 2018), delivering notable achievements in various research fields including molecular property prediction (Jumper et al., 2021; Yang et al., 2022), recommendation systems (Wu et al., 2022b; Gao et al., 2022), and traffic flow forecasting (Liang et al., 2018; Zhou et al., 2020). However, as real-world data is evolving with complex patterns, the challenge of data distribution shift has become a major obstacle for GNNs (Gui et al., 2022; Ji et al., 2022; Wang et al., 2023; Zhou et al., 2022b; Zou et al., 2023; Sun et al., 2022c; 2024). Therefore, various studies concentrate on improving the Out-Of-Distribution (OOD) generalization ability of graph learning models (Chen et al., 2024; 2022; Gui et al., 2024; Miao et al., 2022; Sui et al., 2022; Li et al., 2022; Wu et al., 2022c; Sun et al., 2022b).

Recently, environment-centered invariant learning methods achieved impressive OOD generalization performance with the aim of inferring underlying environment factors in data (Chen et al., 2024; Gui et al., 2024; Xia et al., 2023; Yuan et al., 2023). Those efforts demonstrate that the changes of environment are the fundamental reason for the shift of data distribution (Grice & White, 1961; Liu et al., 2021; Peters et al., 2016). However, existing approaches still lack the ability to decouple causal subgraphs from complex environments. As shown in Fig. 1(a), we double the scale of spurious substructures in the *SPURIOUS-MOTIF(0.5)*, and observe a significant decrease in the performance of current methods when they are re-conducted on this modified dataset. The reason lies in that current methods, even those claiming to model environments, focus much of their attention on extracting causal subgraphs (Chen et al., 2022; Wu et al., 2022a;c; Sun et al., 2021). This results in the model being able to extract causal subgraphs only in known environments, and failing in modeling unseen complex environments. Therefore, this poses a challenging research question: *how to broaden the inference space of environments, enabling model to handle complex environment shifts.*

We argue that this limitation arises from the positive learning paradigm that focuses solely on extracting causal subgraphs as its primary objective. In contrast, negative inference paradigm, modeling the sample space except the invariant

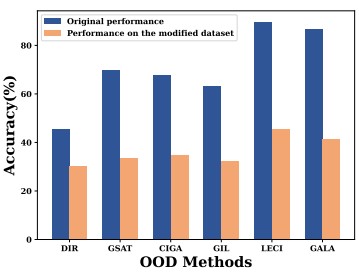

(a) Degradation in performance.

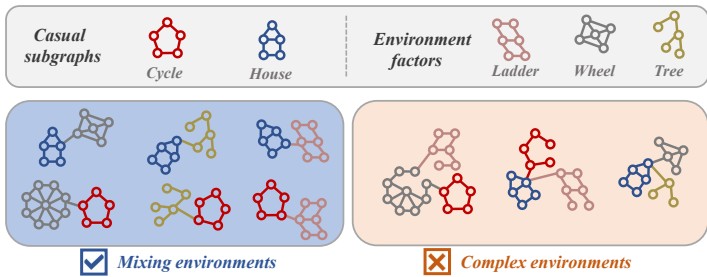

(b) Prediction failure in complex environments.

*Figure 1.* The motivation of our work. (a) We double the scale of spurious substructures in the *SPURIOUS-MOTIF(0.5)* (Ying et al., 2019), and observe a significant decrease in the performance of current methods when they are re-conducted on this modified dataset. (b) The OOD methods, which treat spurious subgraphs as the environments, fail to address the shift of complex environments.

subgraph as environments, has the potential to broaden the perception scope of environment. As shown in Fig. 1(b), the positive inference can only infer the specific *ladder*, *wheel*, and *tree* as environment variables, while the negative inference approach can model all variable space except the *cycle* and *house* as environments. However, the inaccessibility of environment information pose challenges to implementing such negative inference. Specifically, *(1) how to formulate the negative inference learning to achieve environment awareness*, and *(2) how to fully utilize environment information for facilitating causal invariant learning*.

In this work, we propose a novel **Ne**gative inference **G**raph **OOD** framework (NeGo). NeGo aims to achieve causal invariant learning against complex environment shifts by a negative inference. **Firstly**, we design a negative prompt learning framework for inferring underlying environment factors. Given a specific sample with label $Y$, we model all extra-class samples as the environment space for this sample. This design enables the model to capture a broader scale of environments, no more limiting to in-sample spurious subgraphs. **Secondly**, we introduce an environment-enhanced invariant learning strategy to effectively utilize inferred environment variables. Specifically, we design an interactive decoding scheme that utilizes an attention-based residual connection to encapsulate environment embedding into node representations. Different from traditional approaches that neglect the information of environment variables during subgraph extraction (Chen et al., 2024; Gui et al., 2024), our design incorporates the underlying environment patterns into the process of invariant subgraph learning. **Lastly**, we conduct a comprehensive evaluation of NeGo on real-world datasets across domains, and synthetic datasets. NeGo outperforms baselines on nearly all datasets. Our **contributions** can be summarized as follows:

- We observe that existing environment-centered OOD practices encounter difficulties in handling complex environment shifts. Through a comprehensive investigation, we

identify that limited environment space for positive inference is the main reason restricting the generalization capacity of existing OOD approaches.

- We propose a novel invariant learning framework with negative inference NeGo. To be specific, we design an innovative environment inference strategy via negative inference, which effectively broadens the inference space of environment factors. Moreover, we introduce an attention-based residual connection to offer our model with the ability to resist complex environment shifts.

- We conduct extensive experiments on both synthetic and real-world datasets with distribution shifts to evaluate the performance of NeGo. The results from both visualization and quantitative analysis indicate that our framework successfully achieves accurate prediction in complex environmental scenarios.

## 2. Background

**Preliminaries.** A graph is denoted as $G = (\mathcal{X}, \mathcal{A}) \in \mathcal{G}$, where $\mathcal{G}$ is the observed graph dataset. $\mathcal{A} \in \mathbb{R}^{N \times N}$ represents the adjacency matrix and $\mathcal{X} \in \mathbb{R}^{N \times d}$ denotes node features, where $N$ indicates the number of nodes and $d$ is the feature dimension. Each graph is associated with a corresponding label $Y$. From the perspective of causal theory, the graph data can be partitioned into a spurious subgraph $G_S$ and a causal subgraph $G_C$, where $G_C$ directly determines its label $Y$. The spurious subgraph $G_S$ is controlled by the spurious variable $C$, while the causal subgraphs $G_C$ is controlled by the causal invariant factor $C$, as shown in Fig. 4. Based on the different interdependencies among $C$, $S$ and $Y$, structural causal models (SCMs) can be further classified into *Full Informative Invariant Features (FIIF)* and *Partially Informative Invariant Features (PIIF)* (Ahuja et al., 2021; Chen et al., 2022).

**Problem definition.** Our work aims to address the limitations of existing approaches in handling complex data

distribution shifts. We specifically focus on broadening the inference scope of environments, enabling the network to handle intricate scenarios of environment shifts. Additionally, our framework is required to effectively tackle both FIIF and PIIF challenges.

**Environment inference with negative prompt.** Our negative prompter is proposed to achieve a broader inference scale of environments, which is inspired by the success of prompt learning in language models (Brown et al., 2020; Gao et al., 2020). Prompt learning is designed to capture underlying semantic knowledge in language data, which improves the generalization ability of models by introducing appropriate prompt tokens to guide the network learn desired answers (Rao et al., 2022; Sordoni et al., 2024; White et al., 2023; Sun et al., 2023). For example, in the semantic emotion classification task, the language model constructs a template such as "the emotion expressed by this sentence is [class]", where [class] is trained to learn real label. In a similar way, our framework can be viewed as constructing a set of text prompts such as "the underlying environments of current sample are [answer]", where [answer] can be guided to capture the real environment states.

Different from random data augmentation techniques (Han et al., 2022; Li et al., 2021; Lu et al., 2024; Rong et al., 2019; Wang et al., 2021; You et al., 2020; Zhao et al., 2021) and distributionally robust optimization (DRO) methods (Staib & Jegelka, 2019; Wu et al., 2024; Zhu et al., 2021), our prompt-based approach not only broadens the scale of environment inference but also deepens the understanding of underlying data generation process. Existing methods always expand the inference boundary of the model by incorporating stochastic perturbations. However, the introduction of randomness prevents the model from capturing the underlying semantics and hinders its ability to deepen the understanding of generation process. In contrast, our prompt-based approach allows us to deeply extract the underlying casual correlation of variables, which is the reason why we introduce the technique of prompt learning.

## 3. Related works

### 3.1. OOD Generalization.

Out-of-Distribution (OOD) generalization refers to the task of adapting a model that has been trained on a specific distribution to effectively process data from a potentially different distribution. Various approaches can be employed for OOD generalization, including data augmentation (Rong et al., 2019; Wang et al., 2021; You et al., 2020), domain adaptation (Wang & Deng, 2018), and causal invariant learning (Sui et al., 2022; Wu et al., 2022c). Among them, causal invariant learning demonstrates impressive performance in

various fields, due to its powerful interpretability (Chen et al., 2022; Li et al., 2022; Miao et al., 2022; Wu et al., 2022c). Most existing invariant learning methods focus on extracting the causal subgraph to achieve invariant learning. This strategy limits the inference space of the environments to the dimension of spurious subgraphs, which hinders the ability of models to capture the complex environment states. In this work, we propose an invariant learning mechanism based on negative inference to address this limitation.

### 3.2. Prompt Learning

Prompt learning is proposed in NLP models to infer underlying semantic and potential causal associations in linguistic data. Many effective prompt methods have developed with the introduction of large language models, including some hand-crafted prompts (Brown et al., 2020), discrete prompts (Gao et al., 2020; Shin et al., 2020), and learnable prompts design (Li & Liang, 2021). In recent years, prompt learning has also been developed in the graph learning field (Sun et al., 2022a; Li et al., 2024; Sun et al., 2023; Zi et al., 2024). Our approach is the pioneering effort to apply prompt learning to address the graph OOD generalization issue.

### 3.3. Comparisons to Existing Graph OOD Works

Environment-centered studies (Chen et al., 2024; Gui et al., 2024; Li et al., 2022; Wu et al., 2022a; Yang et al., 2022) consider that the data distribution shifts stem from the changes of environments. Therefore, these practices enable the model to withstand data distribution shifts by inferring environment variables. Concretely, the networks are often trained with the objective of equipping models to effectively handle mixed environments scenarios. However, this design allows the networks to make narrow inference about the environments, and makes the networks unable to handle with distribution shifts in complex environments. We attribute this limitation of inference scale to the shortcomings of positive inference, which is proved both empirically and theoretically. To this end, we propose a negative inference mechanism to broaden the inference space for environments, without relying on the mixed environments hypothesis. Our approach, which represents a pioneering practice in utilizing negative inference, is distinct from all existing practices in this field.

## 4. Graph OOD generalization via environment inference

Existing environment-centered practices aim to enable the networks with the ability to resist data distribution shifts. However, our empirical observations indicate that these approaches are insufficient in handling complex environment shifts. To address this issue, we conduct a theoretical

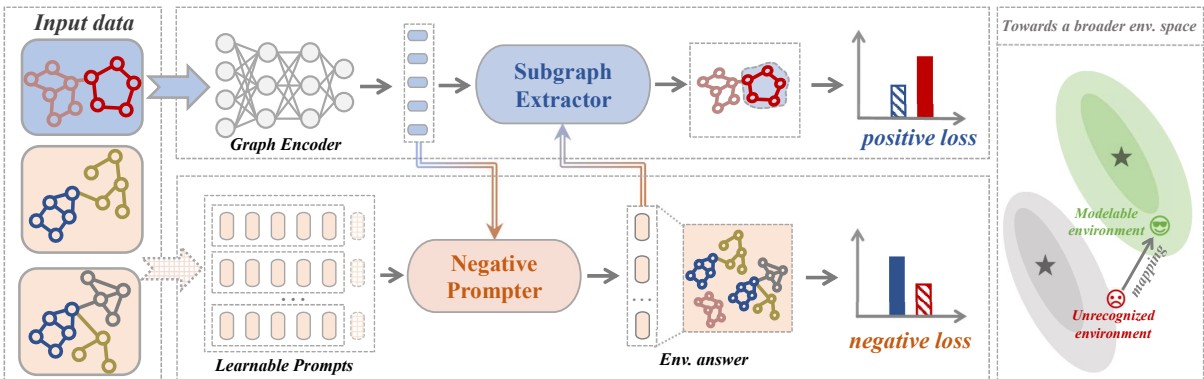

*Figure 2.* The architecture of NeGo. We implements an environment-enhanced graph learning framework in which the environment is extracted through a negative prompt mechanism. The training process is guided by both a positive loss and a negative loss, aiming to broaden the modeling space for the environment.

analysis of these methods, and identify that their failures stem from the limited environment inference space by only positively treating spurious subgraphs as environments. In contrast, we propose a promising method based on negative inference.

### 4.1. Limited Environment Cognitive Space for Positive Inference

From the perspective of causal theory (Pearl, 2009; 2010), the variables of generating the graph data include causal subgraph $G_C$ and spurious subgraph $G_S$, where $G_S$ is controlled by environment variable $E$. As shown in Fig. 4, $G_C \rightarrow Y$ demonstrates a stable casual relationship from $G_C$ to $Y$ in the data generation process. Consequently, the distribution shift between the training data and the test data can be attributed to the shifts of environment $E$, which can be formally expressed as $\mathbb{P}_{train}(\mathcal{G}, E) \neq \mathbb{P}_{test}(\mathcal{G}, E)$. Modeling environment variables becomes crucial for tackling OOD generalization issue (Chen et al., 2024; Gui et al., 2024; Xia et al., 2023; Yuan et al., 2023). With the observed training dataset $\mathcal{G}$, environment-centered approaches strive to learn the distribution of the environment factor $E$,

$$\mathbb{P}(E|\mathcal{G}) = \frac{\mathbb{P}(\mathcal{G}, E)}{\mathbb{P}(\mathcal{G})} = \frac{\mathbb{P}(\mathcal{G}|E)\mathbb{P}(E)}{\int_E \mathbb{P}(\mathcal{G}|E)\mathbb{P}(E)dE}. \quad (1)$$

The prior distribution $\mathbb{P}(E)$ and the likelihood $\mathbb{P}(\mathcal{G}|E) = \prod_{i=1}^{N} \mathbb{P}(G_i|E)$ make the numerator theoretically computable. However, due to the uncertainty in the scale of environments $E$, the denominator of Eq. 1 involving integration becomes intractable. To tackle this issue, we impose a distribution shift boundary based on following environment mixture assumption.

**Assumption 4.1.** (Li et al., 2022) If $K$ different environment labels can be extracted from the observed dataset $\mathcal{G}$,

they are formulated by $K$ independent $D$-dimensional Gaussian distributions $\mathcal{N}(\mu_i, I)$, where $\mu_i \in \mathbb{R}^{1 \times D}$. Therefore, environment variables can be modeled from a vector space perspective, allowing us to approximate the environment space by exploring the mixture space of vectors.

Given the Assumption 4.1, we can model the environments codebook $\boldsymbol{\mu} = (\mu_1, \mu_2, ..., \mu_K) \in \mathbb{R}^{K \times D}$. This environment codebook serves as a proxy for the environment space, representing the entire environment space through the mixture of vectors.

Therefore, the scale of environments is modeled as a mixing space of extracted environment variables. As a result, the new data $G_i$ is associated with the environment state $E_i \sim \mathcal{N}(e_i \cdot \boldsymbol{\mu}, I)$, where $e_i \in \mathbb{R}^{1 \times K}$ representing the mixing weight. This indicates that the latent variables $\boldsymbol{e} = (e_1, e_2, ..., e_N) \in \mathbb{R}^{N \times K}$ be regarded as the proxy factor for the environment variable $E$, directly determining the observed data $\mathcal{G}$ generation process. The posterior probability of the environments $\mathbb{P}(E|\mathcal{G})$ is transformed into,

$$\mathbb{P}(\boldsymbol{e}|\mathcal{G}) = \frac{\mathbb{P}(\mathcal{G}, \boldsymbol{e})}{\mathbb{P}(\mathcal{G})} = \frac{\prod_{i=1}^{N} \mathbb{P}(e_i)\mathbb{P}(G_i|e_i)}{\sum_{e} \mathbb{P}(\boldsymbol{e}) \prod_{i=1}^{N} \mathbb{P}(G_i|e_i)} = \frac{\prod_{i=1}^{N} \mathbb{P}(e_i)}{\sum_{e} \mathbb{P}(\boldsymbol{e})}. \quad (2)$$

The finite space of $\boldsymbol{e}$ allows for the approximation strategy to be feasible. However, the limited scale of $\boldsymbol{e}$ may limit the capacity of model to effectively counter complex environment shifts, which is verified by our empirical results. We next delve into the reason contributing to this limitation. We first present a definition of the *basis* and *base environments* of the environment space, similar to the concepts of basis and base vectors in the vector space.

**Assumption 4.2.** *Let $\boldsymbol{E}_b = \{E_1, ..., E_K\}$ be the basis of environment space, and each element $E_i$ within it is referred*

to as the base environment. The linear combination of base environments can completely describe the entire environment space.

Actually, the environment mixing assumption fundamentally relies on the expectation that extracted environment codebook can cover the basis $E_b$. However, we observe that such goal cannot be achieved by existing methods. Given a graph $G$, current environment-centered methods aim to decompose it into causal subgraph $G_C$ and spurious subgraph $G_S$. The spurious subgraph $G_S$ is inferred as the environment variable. Although $G_S$ is controlled by environment factor ($E \rightarrow G_S$), the space of environment changes encompass more than the scale of spurious subgraphs. For example, consider the substructure $G_C$ that is causally associated with one graph-level property $l$, but the variants of such $G_C$ act as environment factors for other properties. Existing methods fail to capture such other properties with invariance.

**Proposition 4.3.** *Given an observed graph dataset $\mathcal{G}$, the inference process, considering $G_S$ as the environment factor, fails to capture the basis $E_b$ that can represent the entire environment space.*

Proposition 4.3 indicates that the mixing of $\boldsymbol{\mu}$ is unable to encompass the entire environment space. Existing environment-centered methods still have a narrow space of the environments, which leads to incapability of network to extract the causal graph from the complex environments. Therefore, the limitations of existing works are attributed to the narrow inference space of the model for environment variables. Detailed proof can be found in Appendix D.1.

### 4.2. The Enhancement of Negative Inference

Negative inference has a major advantage in effectively expanding the cognitive boundary of models. For example, the positive inference can only infer the specific *ladder*, *wheel*, and *tree* as environment variables, as shown in Fig. 1(b), while the negative inference approach can infer all variable space except the *cycle* and *house* as environments. While the ultimate objective is to extract invariant subgraphs, the negative inference mechanism prioritizes inferring the environment space, empowering the model with the capability to adapt to complex environment shifts. From the perspective of information theory, the training objective of negative inference can be formalized as,

$$\max I(E; G_C|\overline{G^Y}) = \max \ I(E; \mathcal{G} - G_C|Y)$$
$$= \max \ I(E; \mathcal{G}|\overline{G^Y}) - I(E; G_C|Y), \quad (3)$$

where $\overline{G^Y}$ denotes the extra-class samples of the graph $G$ which is labeled as $Y$.

**Proposition 4.4.** *The learning objective of negative inference paradigm (Eq. 3) encompasses a broader cogni-*

tive space for environments, with its upper limit being the ground-truth environment distribution.

Proposition 4.4 emphasizes that the negative inference paradigm enables a broader-scale environment inference space by cooperatively modeling both intra-class spurious subgraphs and extra-class samples. Detailed proof can be found in Appendix D.2.

## 5. Graph Invariant Learning with Negative Inference

In this section, we introduce a novel negative inference graph OOD framework `NeGo` to address the limitation of existing efforts in handling complex environments shifts. Specifically, `NeGo` is developed to design a negative inference learning task to capture underlying environments, and leverage inferred environment embeddings to enhance graph invariant learning.

### 5.1. Negative Prompt Learning for Environment Inference

Negative inference focuses on indirectly extracting invariant subgraph by investigating the information beyond the causal factors. This leads to the problem that the space of variables beyond causal information is infinite-dimensional. Given the insight from Theorem 4.4, we decouple the process of modeling the environment through negative inference into two components: the extraction of intra-class spurious subgraphs and the inference of extra-class samples. Modeling spurious subgraphs is relatively straightforward and extensively studied. The crucial challenge lies in achieving a comprehensive understanding of extra-class sample space.

Formally, let the prior distribution of extra-class samples for $G$ be denoted as $\mathbb{P}(\overline{Y})$, where $G$ is with the label $Y$. We introduce a variational estimate of the environment variables denoted as $\mathbb{Q}_\phi(E|G)$ (a.k.a., $f_\phi$), where $\phi$ is the parameterized network. Denoting KL-divergence as $\text{KL}(\cdot||\cdot)$, the training of $\mathbb{Q}_\phi$ is to implement the first term of Eq. 3, which can be formalized as following optimization,

$$\min_\phi \mathbb{E}[\text{KL}(\mathbb{Q}_\phi(E|G))||\mathbb{P}(\overline{G^Y})]. \quad (4)$$

Inspired by the success of prompt learning in capturing underlying semantic and causal associations in large language models (Floridi & Chiriatti, 2020; Sordoni et al., 2024), we introduce a negative prompter to achieve this goal. Specifically, given a sample $G$ belonging to class $l$, the negative prompter treats all extra-class samples as environments. Designing appropriate prompt tokens to guide effective learning is the primary question that needs to be addressed when employing the concept of prompt learning.

Given the proven efficacy of learnable prompts in various

practices, we design class-specific learnable prompt tokens $\boldsymbol{P} = [\boldsymbol{v}^{(1)}, \boldsymbol{v}^{(2)}, ..., \boldsymbol{v}^{(L)}]$, where $\boldsymbol{v}^{(i)} \in \mathbb{R}^{1 \times d}$ and $L$ is the number of classes. The class-specific design manner aims to capture the extra-class sample space for each graph, in order to achieve the objective defined by Eq. 4. The negative prompter $f_\phi(\cdot)$ is guided to learn the prompt answers $\boldsymbol{A}_N \in \mathbb{R}^{L \times d}$ by interacting the encoded graph embedding $\boldsymbol{Z}_G \in \mathbb{R}^{1 \times d}$ and the learnable prompts $\boldsymbol{P}$,

$$\boldsymbol{A}_N = f_\phi(\boldsymbol{Z}_G, \boldsymbol{P}). \tag{5}$$

The negative prompter $f_\phi$ is parameterized by a cross-attention network in Transformer decoder (Vaswani et al., 2017), where $\boldsymbol{Z}_G$ is obtained by a GNN backbone encoder $h_\psi(\cdot)$. For a sample $G$ with label $l$, such negative prompts answers $\boldsymbol{A}_N$ should satisfy the following two properties:

- The prompts answers $\boldsymbol{A}_N$ should produce a *low match* with graphs whose labels are $l$.

- The prompts answers $\boldsymbol{A}_N$ should produce a *high match* with graphs whose labels are not $l$.

With the explanation in the language models, our *negative prompt mechanism* involves designing prompt tokens to learn the desired [answer] of "the underlying environments of current sample are [answer]". These two properties guide $f_\phi(\cdot)$ to learn a positive answer when interacting with each extra-class sample and a negative answer when interacting with each intra-class sample. Therefore, the training objective of our *negative prompt mechanism* can be formulated as,

$$\begin{aligned} \mathcal{L}_{nega} &= \mathbb{E}[\mathrm{KL}(\mathbb{P}(\overline{G^Y})||\mathbb{Q}_\phi(E|G))] \\ &= -\mathbb{E}[\log \mathbb{P}_\phi(\bar{Y}|G, \boldsymbol{P}) - \log \mathbb{P}_\phi(Y|G, \boldsymbol{P})]. \end{aligned} \tag{6}$$

The environment variables we infer are class-specific, in contrast to the global environment factors constructed by previous methods. Our design is intuitively reasonable, as a specific subgraph may be perceived by one class as causal information, while its minor variations are perceived by other classes as environments. Moreover, it is worth noting that we never overlook the inference of the environments (spurious subgraphs) within intra-class samples. Given that the intra-class environments are always intertwined with causal factors, we incorporate the inference of intra-class environment variables into the discovery of the causal subgraph, which is provided in the next subsection.

## 5.2. Environment-enhanced Graph Invariant Learning

While inferring environment variables is a crucial step in understanding the data generation process, the ultimate goal of graph learning is to achieve casual invariant prediction. Thus, the next challenge to address is the disentanglement of

the causal subgraph from environments. Existing methods often neglect the design of a graph-tailored environment exploitation algorithm, which can lead to the failure in extracting causal subgraphs when environment becomes complex (Gui et al., 2024).

We propose an environment-enhanced invariant learning mechanism that leverages perceived latent environment embeddings to achieve the extraction of causal subgraphs with resistance to complex environment disturbances. Different from the negative prompter that investigates the extra-class sample space, we concentrate on the disentanglement of causal invariant substructures within the intra-class samples in this subsection.

Let the marginal distribution of the causal subgraph $G_C$ be $\mathbb{P}(G_C)$. We introduce a variational estimation of the subgraph extraction $\mathbb{Q}_\xi(G_C|G, E)$ (a.k.a., $g_\xi$), where $\xi$ is the parameterized networks. The model can make casual invariant predictions of the label distribution $\mathbb{P}_\theta(Y|G_C)$ (a.k.a., $g_\theta$), only when the causal graph is accurately extracted from complex environments. The learning objective for environment-enhanced graph invariant learning $\mathbb{P}_\theta \circ \mathbb{Q}_\xi(\cdot)$ is to implement the second term of Eq. 3, which can be formalized as following optimization,

$$\min_{\xi,\theta} \mathbb{E}[\mathrm{KL}(\mathbb{Q}_\xi(G_C|G, E)||\mathbb{P}(G_C)) - \log \mathbb{P}_\theta(Y|G_C)]. \tag{7}$$

The environment embedding $\boldsymbol{A}_N \in \mathbb{R}^{L \times d}$ is inferred at the graph level, but the extraction of substructures often requires node-level operations. Therefore, the primary focus of environment-enhanced invariant learning is to propagate the perceived environment embedding $\boldsymbol{A}_N$ to individual nodes. We design an interaction-decoding module $g_{\xi_1}(\cdot)$ to address this issue.

Specifically, $g_{\xi_1}(\cdot)$ consists of three families of learnable parameters, i.e., $W^Q, W^K, W^V \in \mathbb{R}^{d \times d}$. $g_{\xi_1}(\cdot)$ takes the node-level representation $\boldsymbol{Z} \in \mathbb{R}^{N \times d}$ encoded by the GNN encoder $h_\psi(\cdot)$ and the environment embedding $\boldsymbol{A}_N \in \mathbb{R}^{L \times d}$ obtained by negative prompt as inputs. Three hidden state matrices are calculated by,

$$\boldsymbol{Z}^Q = \boldsymbol{Z}W^Q, \ \boldsymbol{A}^K = \boldsymbol{A}_N W^K, \ \boldsymbol{A}^V = \boldsymbol{A}_N W^V. \tag{8}$$

The node embedding with environment information obtained through residual connections is,

$$\boldsymbol{Z}_E = \mathrm{softmax}(\frac{\boldsymbol{Z}^Q(\boldsymbol{A}^K)^T}{\sqrt{d}})\boldsymbol{A}^V + \boldsymbol{Z}. \tag{9}$$

We exploit a subgraph extractor $G_C = g_{\xi_2}(\boldsymbol{Z}_E)$ to realize invariant subgraph discovery. Then, $G_C$ is encoded by $h_\psi(\cdot)$ to obtain the causal representation for prediction. This representation is passed through an MLP layer $g_\theta$ to model the distribution of $Y$. Therefore, the training objective of

environment-enhanced invariant learning is,

$$
\begin{aligned}
\mathcal{L}_{posi} &= -\mathbb{E}[\log \mathbb{P}_{\xi,\theta}(Y|G_C)] \\
&= -\mathbb{E}[\log \mathbb{P}_\theta(Y|G_C) + \log \mathbb{P}_{\xi_1,\xi_2}(G_C|G, \boldsymbol{A}_N)].
\end{aligned} \tag{10}
$$

### 5.3. Optimization Objective and Theoretical Analysis

Our NeGo achieves a graph learning framework with a wider space of environment inference . This is accomplished through two sequential approaches, first focusing on constructing the learning task for negative inference, and then leveraging the environment embeddings obtained from negative inference to enhance graph causal invariant learning. Thus, the training objective of our NeGo is,

$$
\mathcal{L} = \mathcal{L}_{nega} + \lambda \cdot \mathcal{L}_{posi}, \tag{11}
$$

where $\lambda$ is a hyperparameter, which is set to 1 in the implementation. The training process of NeGo is provided in Alg. 1. Note that the two sub-challenges addressed by NeGo are not independent but closely interconnected. The environment negative inference mechanism assists the network in comprehending the distribution shift of data, while the causal invariant learning with environment enhancement empowers the network to accurately extract causal invariant subgraphs even in complex environments. Thus, the former serves as a foundation for the latter. This design reflects the principle that understanding data generation process is crucial to enhance the generalization of models. We also provide theoretical evidence supporting the ability of NeGo to effectively address both *FIIF* and *PIIF* under both cases of $H(G_C|Y) < H(G_S|Y)$ and $H(G_C|Y) > H(G_S|Y)$. Detailed proof can be found in Appendix D.3.

**Theorem 5.1.** *Given the FIIF or PIIF assumptions under both cases when $H(G_C|Y) < H(G_S|Y)$ and $H(G_C|Y) > H(G_S|Y)$, the causal subgraph $G_C$ can be extracted by optimizing Eq. 11.*

## 6. Experiments

We evaluate the effectiveness of NeGo by answering the following questions. **Q1**. Does our approach effectively address the issue unresolved in existing works? **Q2**. Is our framework sufficiently interpretable? **Q3**. Does each component in our NeGo effectively enhance the generalization capacity? **Q4**. Does our framework operate with high efficiency?

### 6.1. Baselines

We choose four representative OOD methods and seven graph-specific OOD approaches for comparison. The representative OOD frameworks we select consist of ERM, IRM (Arjovsky et al., 2019), V-Rex (Krueger et al., 2021), and

---

**Algorithm 1** The training process of NeGo

**Input:** training data $\mathcal{G}$, negative prompts $\boldsymbol{P}$.
**Initial:** the GNN encoder $h_\psi$, the negative prompter $f_\phi$, environment-enhanced invariant learning mechanism $g_\xi$, final predictor $g_\theta$, learnable prompt tokens $\boldsymbol{P}$, the number of epochs $K$.
**for** $i = 1$ **to** $K$ **do**
 $\boldsymbol{Z}_G = h_\psi(G)$
 $\boldsymbol{A}_N = f_\phi(\boldsymbol{Z}_G, \boldsymbol{P})$
 $\boldsymbol{Z}^Q = \boldsymbol{Z}W^Q$, $\boldsymbol{A}^K = \boldsymbol{A}_N W^K$, $\boldsymbol{A}^V = \boldsymbol{A}_N W^V$
 $\boldsymbol{Z}_E = \text{softmax}(\frac{\boldsymbol{Z}^Q(\boldsymbol{A}^K)^T}{\sqrt{d}})\boldsymbol{A}^V$
 $Y = g_\theta(G_C)$, $G_C = g_2(\boldsymbol{Z}_E + \boldsymbol{Z})$
 **Optimizing:**
 $\mathcal{L}_{naga} = \mathbb{E}[\text{KL}(\mathbb{P}(\overline{G^Y})||\mathbb{Q}_\phi(E|G))] = -\mathbb{E}[\log \mathbb{P}_\phi(\bar{Y}|G, \boldsymbol{P}) - \log \mathbb{P}_\phi(Y|G, \boldsymbol{P})]$
 $\mathcal{L}_{posi} = -\mathbb{E}[\log \mathbb{P}_{\xi,\theta}(Y|G_C)] = -\mathbb{E}[\log \mathbb{P}_\theta(Y|G_C) + \log \mathbb{P}_{\xi_1,\xi_2}(G_C|G, \boldsymbol{A}_N)]$
 $\min_{\psi,\phi,\theta,\xi,\boldsymbol{P}} \mathcal{L} = \mathcal{L}_{nega} + \mathcal{L}_{posi}$
**end for**
**Return** $h_\psi$, $f_\phi$, $g_\xi$, $g_\theta$ and $\boldsymbol{P}$

---

IB-IRM (Ahuja et al., 2021). The Empirical Risk Minimization (ERM) baseline is a vanilla GNN with ERM objective, which is trained using the same settings with (Gui et al., 2024). Graph OOD approaches includes DIR (Wu et al., 2022c), GSAT (Miao et al., 2022), CAL (Sui et al., 2022), CIGA (Chen et al., 2022), GIL (Li et al., 2022), LECI (Gui et al., 2024) and GALA (Chen et al., 2024).

### 6.2. Datasets

We adopt two synthetic datasets with distribution shift and six real-world scenario shift datasets from both molecular and social science domains. **Synthetic datasets** include GOOD-Motif (Wu et al., 2022c) and GOOD-CMNIST (Gui et al., 2022). In **molecular property prediction fields**, we select the scaffold and size splits of GOOD-HIV dataset (Gui et al., 2022; Wu et al., 2018) and the assay and size splits of DrugOOD LBAP-core-ic50 dataset (Ji et al., 2022). We also choose two **social sentiment graph datasets** with distribution shifts, including GOOD-SST2 and GOOD-Twitter (Yuan et al., 2022). Detailed statistics on the number of graphs in those datasets are provided in Tab. 6.

### 6.3. Result Comparison and Analysis

We comprehensively evaluate the OOD performance of NeGo on both real-world and synthetic datasets to answer **Q1**. Tab. 1 and 2 present the performance of NeGo on chemical and sentiment graph datasets. Tab. 8 showcases the performance of our framework on two synthetic datasets. Compared to existing methods, our method achieves optimal performance on almost all datasets. Besides, the per-

*Table 1.* The ROC-AUC performance of `NeGo` on four real-world datasets in chemical research field. ID val and OOD val represent the results of OOD test set using the in-distribution and out-of-distribution validation sets, respectively (Gui et al., 2024). The best results are shown in **bold** and the second best results are underlined. * indicates statistical significance against the second-best results.

| Model | GOOD-HIV-scaffold | | GOOD-HIV-size | | DrugOOD-assay | | DrugOOD-size | |
|---|---|---|---|---|---|---|---|---|
| | ID val | OOD val | ID val | OOD val | ID val | OOD val | ID val | OOD val |
| ERM | 69.61±1.32 | 70.37±1.19 | 61.66±2.45 | 57.31±1.06 | 70.03±0.16 | 72.18±0.18 | 62.97±0.26 | 63.29±0.33 |
| IRM | 73.35±2.30 | 70.89±0.29 | 58.52±0.86 | 60.86±2.78 | 71.56±0.32 | 72.69±0.29 | 63.24±0.26 | 63.46±0.23 |
| V-Rex | 71.73±3.51 | 71.18±0.69 | 58.39±1.54 | 60.10±2.09 | 70.22±0.86 | 72.32±0.58 | 63.87±0.42 | 64.11±0.39 |
| IB-IRM | 67.56±2.31 | 66.25±0.93 | 57.45±0.74 | 56.65±1.22 | 69.34±0.48 | 71.32±0.76 | 64.03±0.61 | 64.59±0.70 |
| DIR | 65.84±1.71 | 68.59±3.70 | 59.69±1.59 | 60.85±0.52 | 67.29±0.73 | 69.70±0.65 | 63.85±0.65 | 64.73±0.54 |
| GSAT | 71.55±3.58 | 71.39±1.41 | 60.92±1.00 | 60.61±1.19 | 71.01±0.54 | 72.26±0.45 | 65.12±0.38 | 65.67±0.45 |
| CAL | 73.48±2.64 | 72.38±1.03 | 62.83±1.26 | 62.58±1.04 | 71.89±0.92 | 71.23±1.13 | 63.85±0.49 | 64.22±0.74 |
| CIGA | 66.25±2.89 | 71.47±1.29 | 58.24±3.78 | 62.56±1.76 | 67.68±1.14 | 70.54±0.59 | 64.14±0.66 | 64.83±0.79 |
| GIL | 70.89±1.60 | 70.23±1.23 | 61.74±1.76 | 61.29±1.34 | 70.45±0.89 | 70.73±1.36 | 64.91±0.51 | 65.43±0.64 |
| LECI | 74.04±0.65 | 74.43±1.69 | 64.83±2.59 | 65.44±1.78 | 72.67±0.46 | 73.45±0.17 | **65.93±0.43** | 66.49±0.60 |
| GALA | 73.85±1.10 | 74.02±1.34 | 63.99±1.54 | 64.45±2.26 | 72.83±0.73 | 73.23±0.29 | 65.23±0.72 | 65.84±0.52 |
| NeGo | **75.21±0.73*** | **75.87±1.02** | **65.23±1.74** | **65.92±1.82*** | **73.20±0.18*** | **73.94±0.25*** | 65.49±0.73 | **66.91±0.84*** |

*Table 2.* The accuracy of `NeGo` on two sentiment graph datasets, where * indicates statistical significance against the second-best results.

| Model | GOOD-SST2 | | GOOD-Twitter | |
|---|---|---|---|---|
| | ID val | OOD val | ID val | OOD val |
| ERM | 78.37±2.64 | 80.41±0.69 | 54.93±0.96 | 57.04±1.70 |
| IRM | 79.73±1.45 | 80.17±1.52 | 55.27±1.19 | 57.72±1.03 |
| V-Rex | 79.31±1.40 | 80.33±1.09 | 56.46±0.93 | 56.37±0.76 |
| IB-IRM | 78.93±1.23 | 80.22±0.55 | 54.23±1.21 | 56.73±1.02 |
| DIR | 77.65±0.71 | 81.50±0.55 | 55.32±1.85 | 56.81±0.91 |
| GSAT | 79.25±1.09 | 80.46±0.38 | 55.09±0.66 | 56.07±0.53 |
| CAL | 81.20±1.21 | 82.34±0.67 | 56.77±0.86 | 57.82±0.44 |
| CIGA | 80.37±1.46 | 82.93±0.75 | 57.51±1.36 | 57.19±1.15 |
| GIL | 81.43±1.02 | 83.31±0.50 | 58.21±1.24 | 57.82±1.18 |
| LECI | **82.93±0.22** | 83.44±0.27 | 59.35±1.44 | 59.64±0.15 |
| GALA | 82.60±0.66 | 82.98±0.42 | 59.03±0.65 | 60.45±1.36 |
| NeGo | 82.72±0.51 | **84.16±0.23*** | **60.82±0.22*** | **61.25±0.70*** |

formance of environment-centered OOD methods, such as LECI and GALA, often achieves suboptimal or even optimal results on various datasets. This demonstrates the effectiveness of modeling environment factors in addressing data distribution shifts.

*Table 3.* The performance comparison (ID val) of NeGo on new environment scenarios.

| Model | GSAT | CAL | CIGA | LECI | GALA | NeGo |
|---|---|---|---|---|---|---|
| Acc.(%) | 70.13 | 75.21 | 71.82 | 78.46 | 77.93 | **79.75** |

To futher validate the effectiveness of our NeGo in new environment scenarios (nonlinear environment mixing), we conduct additional experimental discussion. Considering that the environments in SPURIOUS-MOTIF (Ying et al., 2019) include five specific label-irrelevant base subgraphs (wheel, tree, ladder, star, and path), we increase the environment complexity by diversifying these base subgraphs. Specifically, we randomly add connections to the base subgraphs in each sample, totaling 10% of the original edge

count. As shown in Tab. 3, we observe that the advantage of our NeGo on common OOD datasets is persists in complex environments, with a 1.29% performance improvement.

Moreover, we evaluate the performance of our framework in the the complex environments scenario illustrated in Fig. 1(a). `NeGo` achieves 87.34% and 80.29% on the original and adjusted dataset, respectively. There is only a minor decrease in performance, suggesting that our method effectively tackles the limitations encountered by existing methods in handling complex environments.

To answer the **Q2**, we visually represent the causal subgraphs extracted by `NeGo` on the modified dataset in Fig. 1(a). As depicted in Fig. 3, our method consistently extracts the ground-truth subgraph. The visualized results further validate the effectiveness of our proposed negative inference method. By modeling extensive extra-class samples as environments, our approach offers undeniable advantages in handling complex environment shifts.

### 6.4. Ablation Studies

To answer **Q3**, we investigate each component of `NeGo`. Specifically, we conduct ablation studies to explore the effectiveness of negative prompter and interactive decoding component. Tab. 4 shows that the performance drops significantly when there is either no negative prompter or interactive decoding component. NeGo-NoPro refers to the framework that eliminates negative prompter and negative loss, which causes the most performance drop. Therefore, the negative inference mechanism plays a vital role in enhancing the capability of environment perception. This further validates the rationale of our motivation for incorporating negative inference. NeGo-NoEnv indicates that the casual subgraphs are extracted directly using node embedding without integrating inferred environment information.

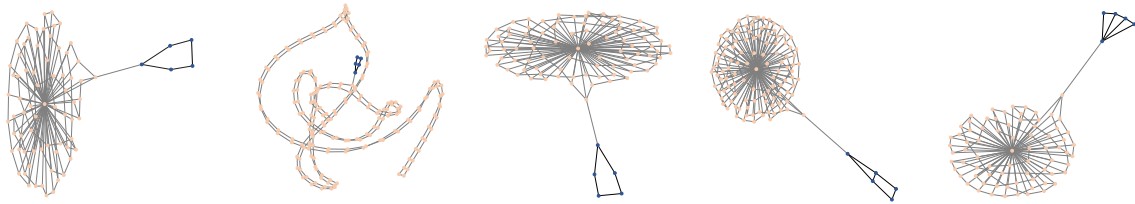

*Figure 3.* The causal subgraphs extracted by NeGo on the modified dataset in Fig. 1(a).

The performance decline emphasizes the significance of environment utilizing strategies overlooked by existing works.

*Table 4.* Ablation studies of NeGo.

| Model | DrugOOD(assay) | GOOD(Twitter) |
|---|---|---|
| NeGo-NoPro | 70.37 | 58.41 |
| NeGo-NoEnv | 71.71 | 59.17 |
| NeGo | **73.20** | **60.82** |

### 6.5. Efficiency Analysis

To address Q4, we explore the training efficiency of NeGo from both theoretical and practical perspectives. The time complexity of NeGo is $\mathcal{O}(|\mathcal{V}| \times d^2 + |\mathcal{V}| \times d \times h + |\mathcal{E}| \times d)$, where $|\mathcal{V}|$ represents the number of nodes, $|\mathcal{E}|$ denotes the number of edges, $d$ is the feature dimension, and $h$ represents the number of cross-attention heads. Our method has linear time complexity with high training efficiency.

We empirically compare the training efficiency of NeGo with other baselines on DrugOOD-size dataset as shown in Tab. 5. Compared with some earlier invariant learning methods (DIR and GSAT), the minor increase in running time of our menthod brings out the substantial performance boost. Additionally, our approach demonstrates greater competitiveness in both training efficiency and performance compared to existing environment-centered methods.

*Table 5.* The training efficiency of NeGo with other baselines on DrugOOD-size (s/epoch).

| Models | GSAT | DIR | CIGA | LECI | GALA | NeGo |
|---|---|---|---|---|---|---|
| Time | 51.6 | 52.6 | 54.2 | 59.1 | 62.3 | 58.7 |

### 7. Conclusion

In this work, we propose a negative inference graph OOD framework NeGo to handle complex environment shift in OOD scenarios. By inheriting the practices of prompt learning in large language models, we design a negative prompter to model the environment on a larger scale. We then introduce an environment-enhanced invariant learning strategy to eliminate spurious subgraphs from the data. This strategy effectively leverages the inferred environment variables to enhance the ability to remove irrelevant information. Extensive experiments on real-world datasets across domains and synthetic datasets validate the effectiveness of NeGo.

## Impact Statement

This paper investigates the advanced application of graph learning in chemistry and social science domains, utilizing publicly available datasets. Consequently, there are no ethical concerns in this work.

## Acknowledgment

This work was supported by the National Natural Science Foundation of China (No.12227901), Natural Science Foundation of Jiangsu Province (BK.20240460, BK.20240461), the grant from State Key Laboratory of Resources and Environmental Information System. The AI-driven experiments, simulations and model training were performed on the robotic AI-Scientist platform of Chinese Academy of Science.

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

# A. More Related works

## A.1. OOD Generalization.

Out-of-Distribution (OOD) generalization learning refers to the task of adapting a model that has been trained on a specific distribution to effectively process data from a potentially different distribution. This study holds significant importance because the issue of data distribution shifts is a common occurrence in the real world. External factors, such as changes in environmental conditions, technological advancements, or evolving user preferences, can lead to shifts in the data distribution. Various approaches can be employed for OOD generalization, including data augmentation (Rong et al., 2019; Wang et al., 2021; You et al., 2020), domain adaptation (Wang & Deng, 2018), and causal invariant learning (Sui et al., 2022; Wu et al., 2022c). Jia et al. (Jia et al., 2024) innovatively proposes a mixup-based environment modeling framework, IGM, to enhance graph invariant learning. IGM focuses on expanding the environment space through a mixing generation scheme, while our NeGo aims to mine environmental space as much as possible from the novel perspective of negative learning. Piao et al. creatively proposes a hierarchical environment inference paradigm to enhance graph invariant learning methods (Piao et al., 2024). This work focuses on generating sample-level hierarchical environments to expand the modeling of the environment space. Unlike this method, our NeGo focuses on class-level environment augmentation, collaborating with extra-class environment modeling and inter-class invariant learning to achieve global inference of environment space.

Among them, causal invariant learning demonstrates impressive performance in various fields, due to its powerful interpretability (Chen et al., 2022; Li et al., 2022; Miao et al., 2022; Wu et al., 2022c). Our NeGo is aligned with this research line, as an environment-centered invariant learning method based on causal theory. However, in the field of graph learning, most existing invariant learning methods focus on extracting the causal graph to achieve invariant learning. This strategy limits the inference space of the environments to the dimension of spurious subgraphs, which hinders the ability of models to capture the complex environment states. In this work, we propose an invariant learning mechanism based on negative inference to address this limitation.

## A.2. Prompt Learning

Prompt learning is proposed in NLP models to infer underlying semantic and potential causal associations in linguistic data. Many effective prompt methods have developed with the introduction of large language models, including some hand-crafted prompts (Brown et al., 2020), discrete prompts (Gao et al., 2020; Shin et al., 2020), and learnable prompts design (Li & Liang, 2021). There have been various works on the interaction of computer vision and natural language processing fields, e.g., text-to-image retrieval text-to-image retrieval (Wang et al., 2019), visual question answering(Antol et al., 2015; Rao et al., 2022; Zhou et al., 2022a) and so on.

In recent years, prompt learning has also been developed in the graph learning field (Sun et al., 2022a; Li et al., 2024; Sun et al., 2023; Zi et al., 2024). Our approach is the pioneering effort to apply prompt learning to address the challenge of graph OOD generalization issue.

## A.3. Comparisons to Existing Graph OOD Works

Environment-centered studies (Chen et al., 2024; Gui et al., 2024; Li et al., 2022; Wu et al., 2022a; Yang et al., 2022) consider that the data distribution shifts stem from the changes of environments. Therefore, these practices enable the model to withstand data distribution shifts by inferring environment variables. Concretely, the networks are often trained with the objective of equipping models to effectively handle mixed environments scenarios. However, this design allows the networks to make narrow inference about the environments, and makes the networks unable to handle with distribution shifts in complex environments. We attribute this limitation of inference scale to the shortcomings of positive inference, which is proved both empirically and theoretically. Therefore, we propose a negative inference mechanism to broaden the inference space for environments, without relying on the mixed environments hypothesis.

Our approach, which represents a pioneering practice in utilizing negative inference, is distinct from all existing practices in this field. DIR (Wu et al., 2022c) aims to identify causal patterns that are stable across different distributions and filter out spurious patterns that are unstable. This work is a classic work in the early application of causal theory to address the challenge of graph OOD generalization. It focuses on obtaining invariant subgraphs with a positive inference manner. GIL (Li et al., 2022) aims to capture the invariant relationships between predictive graph structural information and labels in a mixture of latent environments through jointly optimizing three mutually promoting modules. This method relies on the mixing environment hypothesis and has limited inference space for environments. CIGA (Chen et al., 2022) build

three Structural Causal Models (SCMs) to characterize the distribution shifts that could happen on graphs: one is to model the graph generation process, and the other two are to model two possible interactions between invariant and spurious features during the graph generation, i.e., FIIF and PIIF. This work provides a fresh perspective on existing research on out-of-distribution generalization based on causality. However, it still falls within the framework of positive inference, aiming to extract causal subgraphs. GALA (Chen et al., 2024) utilized proxy prediction mechanism to infer environment label. It is worth noting that the negative samples mentioned in this work are different from our negative inference, and their design is also to improve performance under the mixed environments hypothesis. Thus, it essentially follows a positive inferring process for environment variables. LECI (Gui et al., 2024) primarily focused on spurious substructures space to model the environment variables. Such environment inference strategy still relies on a positive inference with narrow cognitive space of the environments. Unlike existing graph OOD researches that centers on environment awareness, our work presents greater technical challenges, and targets scenarios that are more complex. G-Splice (Li et al.) focuses on linear mixed scenarios with spurious structures, whereas our work aims to address the effectiveness of models in nonlinear mixed scenarios. AIA (Sui et al., 2024) aims to model the environment factors in graph through the lens of data augmentation. Unlike AIA that focuses only on modifications within spurious subgraphs inside samples, our NeGo is more concerned with environment extraction in mixed scenarios at the class level. Thus, in scenarios where the spurious subgraphs within a sample are highly complex, we believe that our method is more effective.

## B. Detailed Datasets

- **GOOD-Motif** (Wu et al., 2022c) is a synthetic dataset designed for studying structure shifts. Each graph in the dataset is created by connecting a base graph and a motif, where the label is determined by the motif. This accessible ground-truth substructure brings a lot of convenience to the invariant subgraph learning with interpretability. This dataset include five label-irrelevant base graphs (wheel, tree, ladder, star, and path) and three label-determining motifs (house, cycle, and crane) are used to generate the graphs in the dataset. In environment-centered invariant learning, such base graphs can be seen as environment factors and such motifs are be consider as the casual factors.

- **GOOD-CMNIST** (Gui et al., 2022) is a semi-synthetic dataset that has been purposefully created to evaluate node feature shifts. It comprises graphs constructed from hand-written digits extracted from the MNIST database, with the transformation applied using superpixel techniques (Monti et al., 2017).

- **GOOD-HIV** (Gui et al., 2022) is a compact and real-world molecular dataset that has been derived from (Wu et al., 2018). It comprises molecular graphs, where atoms represent nodes and chemical bonds represent edges. The primary task associated with this dataset is to predict a molecule's potential for inhibiting HIV replication. Its distribution shift scenario is developed into two, i.e., the scaffold, and the size of nodes in a molecular graph.

- **DrugOOD(LBAP-core-ic50)** (Ji et al., 2022) is utilized in the Ligand-based Affinity Prediction (LBAP) task, where the core noise level and IC50 measurement type serve as domain features. Its distribution shift scenario is developed into three, i.e., the scaffold, the size, and the assay.

- **GOOD-SST2** (Yuan et al., 2022) is a real-world social sentiment dataset derived from natural language. This dataset represents each sentence as a graph, where individual words are treated as nodes, and their corresponding word embeddings serve as node features. The primary task in this dataset involves binary classification, aiming to predict the sentiment polarity of each sentence.

- **GOOD-Twitter** (Yuan et al., 2022) is a real-world natural language sentiment dataset that shares the same transformation process as the SST2 dataset. The classification task of this dataset involves predicting one of three sentiment polarities for each sentence. Similar to the GOOD-SST2 dataset, the sentence lengths are chosen as the domains.

## C. Detailed Baselines

- **DIR** (Wu et al., 2022c) is an early work using causal theory to address the distribution shifts issue in graph data. This work provides detailed theoretical proofs that demonstrate the feasibility of extracting invariant subgraph from graph data.

- **GSAT** (Miao et al., 2022) employ information bottleneck theory to select causal subgraphs under onlythe FIIF assumption. The proposed stochastic attention mechanism in this paper is highly robust in extracting casual subgraphs,

Table 6. Statistics on the number of graphs in the datasets.

| Dataset | Training | ID validation | ID test | OOD validation | OOD test |
|---|---|---|---|---|---|
| GOOD-HIV-Scaffold | 24682 | 4112 | 4112 | 4113 | 4108 |
| GOOD-HIV-Size | 26169 | 4112 | 4112 | 2773 | 3961 |
| GOOD-SST2-Length | 24744 | 5301 | 5301 | 17206 | 17490 |
| GOOD-Twitter-Length | 2590 | 554 | 554 | 1785 | 1457 |
| GOOD-CMNIST-Color | 42000 | 7000 | 7000 | 7000 | 7000 |
| GOOD-Motif-Basis | 18000 | 3000 | 3000 | 3000 | 3000 |
| GOOD-Motif-Size | 18000 | 3000 | 3000 | 3000 | 3000 |
| DrugOOD-assay | 34179 | 11314 | 11683 | 19028 | 19032 |
| DrugOOD-size | 36597 | 12153 | 12411 | 17660 | 16415 |

Table 7. Comparison of existing methods on addressing OOD generalization issue.

| Methods | SCMs | Inferred Environment Space |
|---|---|---|
| DIR | FIIF | Spurious subgraphs |
| GSAT | FIIF | Spurious subgraphs |
| CIGA | FIIF & PIIF | Spurious subgraphs |
| GALA | FIIF & PIIF | Spurious subgraphs |
| LECI | FIIF & PIIF | Spurious subgraphs |
| NeGo | FIIF & PIIF | Intra-class spurious subgraphs and extra-class sample space |

and has emerged as a backbone model in numerous methods. Actually, the subgraph extractor used in our work is also inspired by GSAT.

- **CAL** (Sui et al., 2022) is guided by the backdoor adjustment principle derived from causal theory. It encourages the Graph Neural Networks (GNNs) to focus on exploiting causal features while disregarding shortcut connections.

- **CIGA** (Chen et al., 2022) is the first graph OOD method considering both Fully Informative Invariant Feature (FIIF) and Partially Informative Invariant Feature (PIIF) assumptions. This work presents an OOD algorithm for graphs that is provably generalizable under different types of distribution shifts.

- **GIL** (Li et al., 2022) is designed to capture invariant graph patterns in a mixture of underlying environments and handle the distribution shift issue. This work introduces a GNN-based subgraph generator to identify potentially invariant subgraphs from the complex interaction between invariant and variant patterns.

- **LECI** (Gui et al., 2024) comprehensively reviews existing OOD approaches and identifies the current causalsubgraph discovery challenges. This work jointly optimize label and environment causal independence to achieve powerful causal subgraphs learning.

- **GALA** (Chen et al., 2024) designs an additional assistant model to enhance model with more powerful OOD generalization ability without explicit environment labels. Theoretical proofs establish that GALA possesses robust out-of-distribution generalization capabilities under the FIIF and PIIF assumptions.

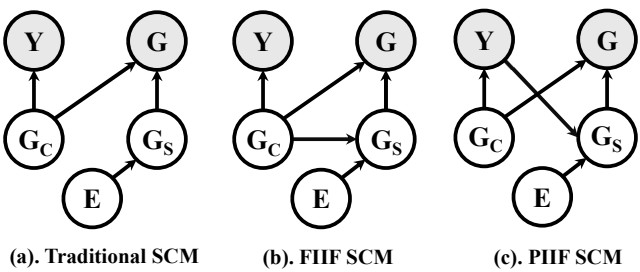

Figure 4. Illustrations of three structural causal models (SCMs).

# D. Detailed Proofs

### D.1. Proof of Proposition 4.3.

The basis $\boldsymbol{E}_b$ represents a set of fundamental components or features that can accurately represent the entire environment space. These components capture the essential variations, patterns, and characteristics present in the environment. However, if the inference process fails to capture this basis, it implies that the process is unable to fully understand and model the complexities of the environment. Thus, we next investigate that whether the environment variable inferred from $G_S$ covers such base environments. We consider two SCMs hypotheses FIIF and PIIF as shown in Fig. 4.

Under the FIIF assumption, $Y \perp G_S | G_C$, we have $P(Y, G_S | G_C) = P(Y|G_C) \cdot P(G_S|G_C)$. This conditional independence assumption leads to an equivalent expression: $P(Y|G) = P(Y|G_S, G_C) = P(Y|G_C)$. Therefore, the process of extracting the causal subgraph $G_C$ is equivalent to the process of modeling the spurious correlations $G_S$. Traditional positive casual learning methods are capable of handling the FIIF assumption.

Under the PIIF assumption, $Y \not\perp G_S | G_C$, we have $P(Y, G_S | G_C) \neq P(Y|G_C) \cdot P(G_S|G_C)$. Furthermore, we can obtain $P(Y|G) = P(Y|G_S, G_C) \neq P(Y|G_C)$. Thus, the process of extracting the causal subgraph $G_C$ cannot be used to infer the labels of samples. More formally, using mutual information theory, we derive the following,

$$I(Y; G_S | G_C) = H(Y|G_C) - H(Y|G_S, G_C) > 0, \tag{12}$$

$$H(Y|G_C) > H(Y|G_S, G_C). \tag{13}$$

This indicates that, given the causal subgraph $G_C$, the uncertainty of $Y$ is higher than when both the spurious subgraph $G_S$ and the causal subgraph $G_C$ are given. It suggests that the spurious subgraph $G_S$ contains additional information of $Y$.

Therefore, the causal subgraph $\hat{G}_C$ learned by the model with the positive learning manner contains components of the spurious subgraph, i.e., $G_S \cap \hat{G}_C \neq \emptyset$. At this point, if we can obtain the basis for the environment space, the model should be able to infer the spurious subgraph $G_S$ and treat it as part of the environment $E$. The extracted causal subgraph $\hat{G}_C$ should be able to effectively remove the spurious subgraph, i.e., $G_S \cap \hat{G}_C = \emptyset$. This clearly contradicts the PIIF assumption, indicating that the model currently lacks the capability to obtain a basis for the environmental space. Therefore, simply inferring the causal subgraph with a postive manner is not sufficient to address the PIIF assumption. Since $E \rightarrow G_S$, modeling the spurious subgraph $G_S$ requires modeling and understanding its root $E$. Existing methods that simply model $G - G_C$ also lack the capability to address the PIIF assumption.

### D.2. Proof of Proposition 4.4.

The optimization of Eq. 3 enables a broader scale environment inference space by cooperatively modeling intra-class spurious subgraphs and extra-class samples. Given that $\max -I(E; G_C|Y) = \max I(E; G_S|Y)$, maximizing $I(E; G_S|Y)$ implements the inference process for intra-class spurious subgraphs. Consider $I(E; \mathcal{G}|\bar{Y}) = \sum_{y_i \in \bar{Y}} I(E; G^{(i)})$, maximizing $I(E; \mathcal{G}|\overline{G^Y})$ implements the modeling of extra-class sample space. The optimization procedure of $\max I(E; \mathcal{G}|\overline{G^Y})$ indicates that all other extra-class samples $\{G^{(i)}|y_i \in \bar{Y}\}$ are modeled as environment variables when making environment inference on samples with label $Y$. Therefore, the optimization process for Eq. 3 encompasses a broader cognitive space for environments, with its upper limit being the ground-truth environment distribution.

### D.3. Proof of Theorem 5.1.

Given that PIIF shifts in the absence of environment labels are more challenging (Chen et al., 2024), our work focuses on the ability of NeGo on the PIIF assumption, namely PIIF implies that the causal variable $G_C$ indirectly influences the spurious variable $G_S$ through the mediator $Y$. In the following analysis, we analyze the two specific scenarios under PIIF assumption, i.e., $H(G_C|Y) < H(G_S|Y)$ and $H(G_C|Y) > H(G_S|Y)$. NeGo aims to comprehensively capture the underlying environment space by inferring the extra-class sample space and the intra-class spurious subgraphs. The learning objective of extracting casual subgraph $G_C$ can be rewritten as follows,

$$\arg \max_{\hat{G}_C \atop \forall e_i, e_j \in E} (I(\hat{G}_C^{e_i}, \hat{G}_C^{e_j}|C) - I(\hat{G}_C, \bar{G}|Y))$$
$$= \arg \max_{\hat{G}_C \atop \forall e_i, e_j \in E} (-I(\hat{G}_C, \bar{G}|Y) + I(\hat{G}_C^{e_i}, \hat{G}_C^{e_j}|Y)), \tag{14}$$

where $\hat{G}_C^{e_i}$ denotes the extracted causal subgraph under any environmental scenario $e_i$. The first term represents the constraint of negative inference, meaning that NeGo models all extra-class samples as environmental space. The second term represents the constraint of positive causal inference, meaning that the causal subgraph extracted under any environmental condition remains consistent, and is most useful for label prediction. Next, we will demonstrate that NeGo can address the two scenarios of the PIIF assumption.

For the case of $I(G_C; Y) > H(G_C) - H(G_S)$, we can get following derivation,

$$H(G_S|Y) > H(G_C|Y), \tag{15}$$

$$H(G_S) - I(G_S; Y) > H(G_C) - I(G_C; Y), \tag{16}$$

$$H(G_S) - H(G_C) + I(G_C; Y) > I(G_S; Y) > 0, \tag{17}$$

$$I(G_C; Y) > H(G_C) - H(G_S). \tag{18}$$

We can get that inferring $G_C$ from $Y$ is more effective and seamless compared to simply separating causal and spurious substructures based on entropy differences. Thus, our positive inference approach, $\arg\max_{\forall e_i, e_j \in E} I(\hat{G}_C^{e_i}, \hat{G}_C^{e_j}|Y)$, is sufficient to achieve the decoupling of $G_C$ from the label $Y$.

For the case of $I(G_C; Y) < H(G_C) - H(G_S)$, we get $I(G_C; Y) < H(G_C) - H(G_S)$. This means that we need to consider entropy differences in the data composition to assess the differences between causal and spurious relationships. In other words, positive inference $\arg\max_{\forall e_i, e_j \in E} I(\hat{G}_C^{e_i}, \hat{G}_C^{e_j}|Y)$ alone may result in $\hat{G}_C$ containing spurious subgraph information, meaning $G_S \in \hat{G}_C$. Fortunately, our negative inference strategy can further refines $\hat{G}_C$ by considering entropy differences $H(G_C) - H(G_S)$ to better distinguish between causal and spurious relationships. Specifically, our $G_C$ is also subject to this constraint through a negative inference approach to learn $\hat{G}_S$,

$$G_C \in G - \arg\max(I(Y|\hat{G}_S) - I(\hat{G}_S|\bar{Y})). \tag{19}$$

## E. More Experiment Results

### E.1. Implementation Details

We implement our Nego and parts of baselines with PyTorch 1.10.1 on a server with NVIDIA A100-PCIE-40GB. All experiments are repeated with 10 different random seeds of [1,2,3,4,5,6,7,8,9,10]. The reported results include the mean and standard deviation obtained from these 10 runs. During the training stage, we employ the Adam optimizer. We set the maximum number of training epochs to 200. The batch size of training is set as 32 except for GOOD-CMNIST, which uses a batch size of 64. For GOOD-Motif, GOOD-CMNIST and GOODSST2, the learning rate is set to $5 \times 10^{-4}$. For GOOD-HIV, GOOD-Twitter, and DrugOOD, we exploit a learning rate of $10^{-4}$. Additionally, we utilize a weight decay of $10^{-4}$ to help with regularization and prevent overfitting. The experiment setup of all baselines is same as (Gui et al., 2024).

### E.2. Performance on the Synthetic Datasets

As shown in Tab. 8, our method achieves optimal performance on almost all datasets. Besides, the performance of environment-centered OOD methods, such as LECI and GALA, often achieves suboptimal or even optimal results on various datasets. This demonstrate the effectiveness of modeling environment factors in addressing data distribution shifts.

### E.3. Hyperparameter Sensitivity Analysis

To investigate the sensitivity of $\lambda$ to performance in Eq. 11, we conducted experiments on both GOOD-HIV-scaffold and DrugOOD-assay. As illustrated by the results from Fig. 5(b), the performance of NeGo improves significantly when $\lambda \approx 1$. We also find that neither smaller nor larger values of $\lambda$ lead to improvements in performance. Therefore, only by jointly utilizing both positive learning and negative inference can better performance be achieved.

*Table 8.* The accuracy of `NeGo` on two synthetic datasets, where GOOD-Motif has a structure shift and GOOD-CMNIST has a feature shift. * indicates statistical significance against the second-best results.

| Model | GOOD-Motif | | GOOD-CMNIST | |
|---|---|---|---|---|
| | basis | size | color | covariate |
| ERM | 60.93±11.11 | 56.63±7.12 | 26.64±2.37 | 57.56±9.59 |
| IRM | 64.94±4.85 | 54.52±3.27 | 29.63±2.06 | 58.11±5.14 |
| V-Rex | 61.59±6.58 | 55.85±9.42 | 27.13±2.90 | 48.78±7.81 |
| IB-IRM | 63.45±5.42 | 52.76±4.67 | 28.95±1.98 | 50.56±6.62 |
| DIR | 34.39±2.02 | 43.11±2.78 | 22.53±2.56 | 44.67±0.00 |
| GSAT | 62.27±8.79 | 50.03±5.71 | 35.02±2.78 | 68.22±7.23 |
| CAL | 59.45±3.34 | 51.27±2.50 | 28.87±1.80 | 52.59±2.76 |
| CIGA | 37.81±2.42 | 51.87±5.15 | 25.06±3.07 | 56.78±2.99 |
| GIL | 68.48±2.46 | 63.61±2.75 | 47.32±2.27 | 57.61±2.98 |
| LECI | **84.56±2.22** | 71.43±1.96 | 51.80±2.71 | **83.20±5.89** |
| GALA | 80.95±1.31 | 70.45±1.30 | 52.68±2.40 | 81.23±3.29 |
| NeGo | 83.96±1.90 | **72.65±1.47**\* | **53.28±1.79** | 82.43±1.73 |

## E.4. Case Studies

We also explore whether incorporating prompt learning can enhance the model's performance, rather than our overall negative prompt framework. To this end, we develop a variant of our NeGo framework, referred to as PoGo, which incorporates the positive prompt practice. We evaluate the effectiveness (ROC-AUC) of PoGo on four distribution shift datasets. We present the final performance by averaging the results from two runs conducted on an NVIDIA A100-PCIE-40GB with different random seeds. As shown in Fig. 5(a), the performance of PoGo is competitive with recent successful practices like LECI and GALA, demonstrating that the design of positive prompt can still obtain excellent generalization. However, our framework of negative prompt shows superior performance.

We further investigate the reason of such performance of positive prompt practice PoGo. We modify PoGo by masking the $\mathcal{L}_{posi}$ (the original Negative Loss $\mathcal{L}_{naga}$), obtaining PoGo (w/o. $\mathcal{L}_{posi}$). With all other configurations remaining the same, we observe a significant decrease in the performance of PoGo (w/o. $\mathcal{L}_{posi}$). Our analysis is as follows: although both $\mathcal{L}_{posi}$ and $\mathcal{L}_{pred}$ are positive losses in PoGo, we argue they serve different purposes and convey distinct information. $\mathcal{L}_{prompt}$, as a guidance strategy for the positive prompt, guides the prompt module to learn more potential environment semantics, while $\mathcal{L}_{pred}$ enhances prediction accuracy. Without prompt guidance $\mathcal{L}_{prompt}$, the advantage of prompt learning is not released. Therefore, we argue that positive prompt may also enhance the model to capture a broader scale of environments. A more in-depth investigation will be left for our future work.

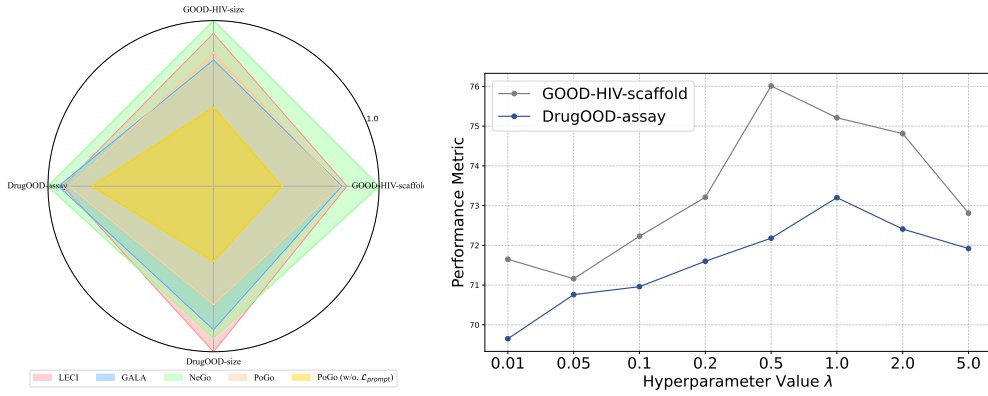

*Figure 5. Left:* the performance comparison of PoGo, which is a variant of our NeGo framework by incorporating the positive prompt practice. *Right:* hyperparameter sensitivity analysis.

## F. Future Works

Our design can effectively solve the existing challenges, but there still exist a limitation. The negative prompter in our approach learns class-specific environment embeddings by considering all extra-class samples as environment variables. This results in our method relying on the class information of the dataset. With a larger number of classes, the model is better equipped to capture and recognize complex underlying environment factors. When the dataset is limited to a binary classification task, environment factors always present within the in-class samples. In this case, our negative prompter may have reduced capability to expand the environment inference space. The reason for this limitation is that the model is sensitive to the characteristics of dataset. Actually, we can realize that environment variables are often shareable across datasets. Therefore, it is a promising research direction to study cross-task graph OOD work to capture broader environmental information. In the future, we aim to investigate transferable multi-task graph OOD generalization learning, which is not discussed in existing works.

