# OpenReview forum: "Enhancing Graph Invariant Learning from a Negative Inference Perspective"
_ICML.cc/2025/Conference — ICML 2025 poster_

### Official Review · Reviewer_vzxf · 2025-03-05

**Overall Recommendation:** 4

**Summary:**

The paper presents NeGo, a negative inference graph OOD architecture designed to address complex environmental shifts in OOD scenarios. Extensive experiments demonstrate the effectiveness of proposed architecture.

**Claims And Evidence:**

Yes, the Yes, this work brings a new perspective and method to graph OOD learning.laims are supported by both theoretical analysis and empirical experiments.

**Essential References Not Discussed:**

The introduction to related work is relatively thorough.

**Experimental Designs Or Analyses:**

Yes, the experiment section is structured around four Questions and provides a comprehensive discussion on the effectiveness of NeGo.

**Methods And Evaluation Criteria:**

Yes, this work brings a new perspective and method to graph OOD learning.

**Other Comments Or Suggestions:**

No Other Comments.

**Other Strengths And Weaknesses:**

Strengths:
1. This manuscript proposes a novel approach to address complex environment shift in OOD scenarios. By introducing an interesting perspective on negative learning, and a novel technique for prompt learning, the authors construct approaches that specifically address existing challenges.

2. Both the technical details and experiments are very well explained and organized. This manuscript is well-written, the technical design is well-articulated, and the motivation is well-stated.

3. The theoretical analysis presented in the paper is sound and rigorous. The reproducibility of the paper is good.
Weaknesses:
1. The technical contributions of this manuscript can be viewed in two aspects: the perspective of negative learning and the utilization of prompt learning. The manuscript lacks an intuitive analysis of their individual roles, such as whether these two elements must be combined jointly, what each contributes when combined.

2. The learnable parameters (learnable prompt) is the input to the negative learning module and serves as a key component. Although this design has been validated by many previous works, the authors should provide a more thorough analysis of its function.

3. Although the proposed framework does not involve excessive intermediate computations, it is necessary to discuss the efficiency of model.

**Questions For Authors:**

1. Is it necessary for negative learning and the utilization of the prompt learning mechanism to be combined, and what does each contribute when they are used together?

2. Why can the learnable prompt guide negative learning? Do the authors offer insights that differ from previous works?

3. What is the training efficiency of the proposed model?

**Relation To Broader Scientific Literature:**

This paper aims to solve the complex environment shift issue that can not be solved by the existing works, which is a further improvement on the existing methods.

**Theoretical Claims:**

Yes, the theoretical proofs in the manuscript are solid.

---

> ### Author Rebuttal · Authors · 2025-03-31
>
> Dear Reviewer vzxf,
>
> Thank you so much for taking your valuable time to review our manuscript. We greatly appreciate the positive feedback on our work. Here we will carefully address your concerns.
>
> **W1&Q1. Intuitive analysis**
>
> Prompt learning and negative learning collaboratively contribute to performance improvement, which is studied in Apendix E.4. Given your valuable comment, we will move the discussion from the Appendix to the main text in the revised version of our manuscript. Specifically, we designed two variants of NeGo: PoGo uses a positive learning mechanism, and PoGo (w/o. prompt) masks the prompt learning module. The performance comparison results we obtained are as follows: NeGo > PoGo > PoGo (w/o. prompt). Therefore, using only negative learning mechanism or only the prompt module does not achieve the maximum performance.
>
>
> **W2&Q2. Learnable parameters**
>
> The reasons for choosing learnable parameters as prompt guidance are as follows:
>
> - Limitations of manual design. Traditional manual prompts require expert knowledge and may not generalize well to new data or tasks. Especially in scenarios with data distribution shifts, manually designed prompts often struggle to generalize across all distributions.
>
> - A data-augmented perspective. The learnable prompt enables implicit enhancement by dynamically (learnably) updating the input prompts without directly altering the original data distribution. This design is particularly well-suited for tasks involving data distribution shifts.
>
>
> **W3&Q3. The efficiency of model**
>
> We support that model efficiency is a crucial evaluation criterion. In the next version of the manuscript, we will incorporate Appendix E.5 and following table into the main text.
>
> | Models   | GSAT | DIR | CIGA | LECI | GALA | NeGo |
> |   :--:   | :--:   |  :--:   |  :--:   |  :--:   |  :--:   |   :--:   |
> | Time(s/epoch) | 51.6 |  52.6  |  54.2 |  59.1 |  62.3 | 58.7 |
>
>
> Thank you again for taking the time to review our work! We will carefully consider your suggestions and further revise our manuscript to satisfy the high standards of ICML community.

---

### Official Review · Reviewer_wCN4 · 2025-03-08

**Overall Recommendation:** 4

**Summary:**

This paper introduces a negative inference graph OOD framework (NeGo) to broaden the inference space for environment factors. Technically, this paper is inspired by the concept of prompt learning and, from the perspective of negative learning, propose an innovative approach called negative prompt. Experimentally, this paper focuses on the graphs with complex environments, and demonstrates the generalization ability of NeGo in complex environments by comparing it with baselines.

## update after rebuttal
I appreciate the authors' response. I will maintain my positive score.

**Claims And Evidence:**

Yes. The claims made in the submission are supported by clear and convincing evidence. The authors provide a solid theoretical foundation, backed by empirical experiments that reinforce their assertions.

**Essential References Not Discussed:**

The authors offer a thorough discussion of the related works, covering nearly all relevant references.

**Experimental Designs Or Analyses:**

Yes. The overall experimental setup, including the choice of datasets and evaluation metrics, seems well thought out and appropriate for this research.

**Methods And Evaluation Criteria:**

Yes, the proposed methods and evaluation criteria are well-suited for the challenge in this field. The chosen baselines align with the goals of the research, and the evaluation criteria is appropriate for assessing the performance and effectiveness of the proposed approach.

**Other Comments Or Suggestions:**

No

**Other Strengths And Weaknesses:**

This paper is a good work with a solid motivation, clear writing, and novel method. Specifically, the strengths of this work excel in following aspects,

- This paper tackles a novel and important issue, with limited prior work on focusing graph OOD challenge under complex environment shifts.

- The authors empirically investigate the performance of existing models in complex environments shifts, effectively verify the significance of the work studied.

- The technical contribution is substantial and innovative. To tackling existing issue, proven effective techniques and insights are successfully applied in this study.

- The writing is clear, and the analyses are compelling.

The paper still has some notable weaknesses. If the following weaknesses are addressed, the paper could be significantly improved:

- The paper includes numerous network architecture designs ($f_\phi}$, $g_\xi $, $g_\theta$), but these parameterized networks are only described in text, making them hard to understand and visualize. For example, in Figure 2, it is difficult to understand at which stage $f_\phi$, $g_\xi $, and $g_\theta$ appear and what roles they play.

- The mechanism of negative inference aims to capture invariant subgraphs by modeling extra-class samples, which is similar with the concept of negative sampling in traditional machine learning. I'm confused about the distinction between these two concepts.

- The abstract section should contain both an introduction to the framework and an evaluation of the effectiveness of the model. But the abstract of this work lacks a quantitative evaluation of the performance of NeGo.

- The experimental comparison of training efficiency should be presented in the main text, not in the appendix.

- Minor typo. line95 “while the causal subgraphs”-> “while the causal subgraph”

**Questions For Authors:**

- Prompt learning has been widely studied in text and image structured data. What unique design or contributions do the authors make when transferring the mechanism of prompt learning to graph-structured data in this work?

- I am confused about why negative prompts are so effective. Could you provide some intuitive validations to offer more insights?

- What is the transferability of the proposed method? Could it inspire insights for other research, such as self-supervised learning?

**Relation To Broader Scientific Literature:**

The key contributions of the paper build upon prior research in graph OOD. For instance, the authors extend the works [1-3], all of which focus on solving the distribution shift issue on graphs. Note that the contributions of this paper are quite novel, addressing a gap in the literature that has not been fully explored before.

[1] Discovering invariant rationales for graph neural networks, ICLR 2022;
[2] Does invariant graph learning via environment augmentation learn invariance? NeurIPS 2023;
[3] Joint learning of label and environment causal independence for graph out-of-distribution generalization, NeurIPS 2023.

**Theoretical Claims:**

Yes. I have reviewed the correctness of the theoretical proofs presented in the manuscript. The proofs for Proposition 4.3., Proposition 4.4. and Theorem 5.1. appear to be sound and logically consistent.

---

> ### Author Rebuttal · Authors · 2025-03-31
>
> Dear Reviewer wCN4,
>
> Thank you for taking valuable time to review this work. We have carefully considered your comments, and the following are our detailed responses. We hope these details could address your concerns.
>
> **W1. Parameterized networks**
>
> ${f_\phi }$ represents the negative prompter to learn negative embedding, ${g_\xi }$ and ${g_\theta }$ are subgraph extractor and final predictor. We will update Fig. 2 by adding the parameter networks to the structure figure in the next version.
>
> **W2. Negative learning and negative sampling**
>
> The differences between our negative learning (NL) and negative sampling (NS) is as follows:
>
> - NS primarily aims to optimize training efficiency, while NL focuses on enhancing the model's generalization and distinguishment ability.
>
> - NS involves randomly selecting a small number of negative samples from all negative samples for training. In contrast, our NL design negative loss to explicitly guide the model in learning how to distinguish between positive and negative samples, and use prompt mechanism to increase the contribution of negative samples to the model's generalization.
>
>
> **W3&4. Quantitative evaluation and Training efficiency**
>
> We will add the following in the description: *Our NeGo achieves optimal performance across four datasets, with the highest performance improvement of 1.17%.* In the next version of the manuscript, we will incorporate Appendix E.5 (Efficiency Analysis) into the main text.
>
>
> **W5. typo**
> Many thanks for your careful review. We will certainly correct the typos you pointed out and proceed to check our manuscript in greater detail.
>
>
> **Q1. Unique design of graph-structured data**
>
> The unique prompt design of our NeGo lies that the prompt outputs serves as a latent embedding for extracting subgraphs, rather than being directly used for prediction. Given that label-related information in graph-structured data corresponds to subgraphs, we employ prompt learning with the goal of disentangling causal subgraph from spurious subgraph.
>
>
> **Q2. Intuitive validations**
>
> The effectiveness of negative prompting in NeGo can be intuitively understood from following several perspectives:
>
> - Broadening the Environment Inference Space. Negative prompting allows the model to consider all extra-class samples as potential environments. This significantly expands the scope of environment factors that the model can infer, beyond just the spurious subgraphs within the same class.
>
> - Our Case Studies and Visualization. In Appendix E.4, we show that the model consistently identifies the ground-truth subgraphs even in complex environments. This demonstrates that negative prompting helps the model accurately distinguish between causal and spurious factors.
>
>
> **Q3. Transferability**
>
>
> - Transferability Across Tasks. The negative inference mechanism of NeGo not only focuses on extracting causal subgraphs but also learns environment factors through negative prompting, a mechanism that may have a certain degree of universality across different tasks. Experimental results show that NeGo performs well in different tasks, such as molecular property prediction (e.g., GOOD-HIV, DrugOOD) and social sentiment analysis (e.g., GOOD-SST2, GOOD-Twitter). This shows that the design of NeGo can accommodate the demands of different tasks and has a certain level of transferability across tasks.
>
> - The design of NeGo is based on causal theory and negative inference, theories that have broad applicability across different fields (such as chemistry, biology, social sciences). Experimental results indicate that NeGo performs well in molecular property prediction tasks in the chemical domain and social sentiment analysis tasks in the social domain. This suggests that the design of NeGo can adapt to data distribution shifts in different fields and has a certain level of transferability across domains.
>
>
> Thanks again for your constructive comments, which are valuable in helping us further improve our work. Look forward to your further feedback!

---

### Official Review · Reviewer_sxw9 · 2025-03-08

**Overall Recommendation:** 3

**Summary:**

In this work, the authors tackle the complex environment shift challenge in graph learning. Through theoretical analysis and an in-depth discussion of the challenge, the authors propose the negative prompt graph learning framework NeGo. Extensive experiments on real-world datasets across domains and synthetic datasets validate the effectiveness of NeGo.

**Claims And Evidence:**

Yes. The claims of this manuscript are supported by clear observations and convincing experiment results.

**Essential References Not Discussed:**

No. Section 3 and Appendix A provide a comprehensive analysis of relevant works, including many classic studies of graph learning, and recent researches of graph OOD learning.

**Experimental Designs Or Analyses:**

Yes. The experimental designs of this manuscript are comprehensive, including comprehensive datasets, a robust set of baselines, and a wealth of experimental results.

**Methods And Evaluation Criteria:**

Yes. The research motivation of this manuscript stems from the limitations of existing works, and an innovative approach is proposed to address these issues. As a result, it contributes to advancing the field.

**Other Comments Or Suggestions:**

There are also a few typos and statements require to be corrected. Line 204: delete ‘first’; Line 268: some references should be added after ‘various practices’; Line 351: ‘supporting’ should be modified to ‘to support’.

**Other Strengths And Weaknesses:**

The strengths I found are listed as below,
S1. The motivation of this research is significant. Section I presents many intuitive examples and descriptions of existing work, effectively helping reviewers and readers grasp the focus of this research.

S2. The background is comprehensive, including prompt learning and two distribution shift hypotheses. Besides, this paper provides a clear analysis of the differences between the proposed method and existing works.

S3. The experiment design in this paper is thorough. The four Questions preset in the experiment can fully prove that the proposed method is effective in tackling existing issues. In the comparison of model performance, it is convincing that the authors used the significance hypothesis for validation.

S4. The reproducibility of this research is trustworthy, and the technical implementation is provided.
The weaknesses are listed as follows,
W1. Environmental variables in OOD scenarios typically refer to external factors that influence model performance and data distribution, which may differ from the distribution of the training data. Are the environment variables in graph-structured data limited solely to the size of the graph, or do they also encompass other factors such as node attributes, edge relationships, or external contextual information that may influence the graph's behavior?

W2. In the experiment section, I observe that the suboptimal performance of NeGo is often concentrated in the 'ID val' setting. This suggests that the same model tends to exhibit different preferences under 'ID val' and 'OOD val' conditions. The authors do not provide a thorough discussion of this phenomenon.

W3. The paper provides extensive theoretical Proof to verify NeGo can handle both the FIIF and PIIF assumptions. What is the significance of distinguishing FIIF and PIIF two assumptions?

**Questions For Authors:**

Please answer the Weaknesses (W1-W4).

**Relation To Broader Scientific Literature:**

The topic explored in this manuscript is one that has been extensively studied in recent scientific literatures. As a result, the proposed framework can offer novel ideas (negative prompt) to advance scientific research in the field.

**Theoretical Claims:**

Yes. The theoretical claims of this manuscript are proved in the Appendix.

---

> ### Author Rebuttal · Authors · 2025-03-31
>
> Dear Reviewer sxw9,
>
> Thank you for taking valuable time to review our work. We have carefully considered your comments, and the following are our detailed responses. We hope these details could address your concerns.
>
>
> **W1. The environment in the graph data.**
>
> The environment factors in graph data are highly informative and extend well beyond just graph scale. In graph-structured data, environment variables are typically defined as all information (spurious subgraph) beyond the causal subgraph. In synthetic datasets, researchers commonly simulate environment variations by increasing the size of spurious subgraph. For example, in GOOD-Motif dataset, base graphs (label-irrelevant subgraphs) can be seen as environment factors and  motifs (label-determining subgraphs) are be consider as the casual factors. In real molecular datasets, the environment variables are often characterized by identifying molecules with complex structure and a diverse range of atomic types.
>
> Actually, the complex environments discussed in our manuscript encompass not only changes in scale but also variations in subgraph types. The environment variables for each dataset are detailed in the table below.
>
>
> |      | Environment factor |
> |   :--:   | :--:   |
> | GOOD-Motif |  Label-irrelevant base graphs |
> | GOOD-CMNIST | Color |
> | GOOD-HIV |  The scaffold and the size  |
> | DrugOOD(LBAP-core-ic50) | The scaffold, the size, and the assay |
> | GOOD-SST2  | Sentence lengths |
> | GOOD-Twitter   | Sentence lengths |
>
>
> **W2. A discussion of experiment phenomenon.**
>
>
> Thank you for your insightful comment. In real-world tasks, causal variables extend beyond the subgraph itself and also incorporate additional contextual information. We aruge that negative learning may cause the model to actively suppress certain features that are beneficial for ID scenarios but ineffective in OOD scenarios. The phenomenon where our NeGo achieves suboptimal performance on ID val but delivers the best results on OOD val validates our view.
>
> **W3. FIIF and PIIF.**
>
> FIIF and PIIF represent two possible interactions between invariant and spurious features during graph generation, as modeled by SCMs [1]. FIIF and PIIF have been accepted by many studies for verifying whether the proposed model demonstrates comprehensive invariant learning ability [2,3].
>
> Under the FIIF assumption, the invariant feature $C$ fully contains all the information about the label $Y$, i.e., $(S, E) \perp Y \mid C $. This means that given $C $, $S$ and $E $ are independent of $Y$. In the structural causal model under FIIF, we have $Y := f_{\text{inv}}(C)$, $S := f_{\text{spu}}(C, E)$, $G := f_{\text{gen}}(C, S)$. Here, $f_{\text{inv}}$ is the label generation function, $f_{\text{spu}}$ describes how $S$ is influenced by $C$ and $E$, and $f_{\text{gen}}$ is the graph generation function. Under the PIIF assumption, the invariant feature $C$ only partially contains the information about the label $Y$, i.e., $S$ is indirectly dependent on $C$ through $Y$. In the structural causal model under PIIF, we have $Y := f_{\text{inv}}(C)$, $S := f_{\text{spu}}(Y, E)$, and $G := f_{\text{gen}}(C, S)$. Here, $f_{\text{inv}}$ is the label generation function, $f_{\text{spu}}$ describes how $S$  is influenced by $Y$ and $E$, and  $f_{\text{gen}}$ is the graph generation function.
>
> In both FIIF and PIIF, the interaction between $C$ and $S$ is different, leading to distinct distribution shift patterns. In FIIF,  $C$ directly controls  $S$ , while in PIIF, $C$ indirectly controls $S$ through $Y$ . Understanding these differences is crucial for addressing distribution shifts. In our work, we also validate that our NeGo possesses comprehensive causal learning ability through theoretical proof. In the next version of our manuscript, we will introduce more discussions on the importance of studying causal learning under the FIIF and PIIF assumptions.
>
>
> **Typos.**
>
> Many thanks for your careful review. We will certainly correct the typos you pointed out and proceed to check our manuscript in greater detail. We will carefully consider your comments and revise our manuscript to improve it further. Looking forward to your feedback!
>
> **References:**
>
> [1] Chen, Yongqiang, et al. "Learning causally invariant representations for out-of-distribution generalization on graphs." Advances in Neural Information Processing Systems 35 (2022): 22131-22148.
>
> [2] Chen, Yongqiang, et al. "Does invariant graph learning via environment augmentation learn invariance?." Advances in Neural Information Processing Systems 36 (2023): 71486-71519.
>
> [3] Gui, Shurui, et al. "Joint learning of label and environment causal independence for graph out-of-distribution generalization." Advances in Neural Information Processing Systems 36 (2023): 3945-3978.

---

### Official Review · Reviewer_cMu6 · 2025-03-12

**Overall Recommendation:** 4

**Summary:**

Through the experimental exploration, this paper observe that the existing approaches lack the ability to decouple causal subgraphs from complex environments. To tackle this problem, a negative prompt environment inference framework is proposed. The theoretical analysis, along with the excellent experimental performance, effectively demonstrates the superiority of the proposed method.

**Claims And Evidence:**

Yes. All claims made in the submission supported by clear and convincing evidence.

**Essential References Not Discussed:**

The works discussed in this literature is very fresh and representative.

**Experimental Designs Or Analyses:**

Yes, I have checked the validity of all experimental designs. It is better to provide more interpretable analysis.

**Methods And Evaluation Criteria:**

Yes. The proposed method and evaluation criteria make sense for the problem at hand.

**Other Comments Or Suggestions:**

No Other Comments Or Suggestions.

**Other Strengths And Weaknesses:**

Strengths:

1. The topic on studying graph earning methods in the scenario of complex environments shift is important to the community. The authors provide new insights to show the relationship between positive and negative learning.

2. The paper’s core contribution, a negative inference graph OOD framework (NeGo), is thoroughly tested and demonstrates strong empirical performance. This paper compares the performance of NeGo with fresh baselines, and conducts thorough discussions under the cases of complex environment shifts.

3. The presentation of the paper is clear. The overall logical structure of the paper is clear and easy to follow.

Weaknesses:

1. While the paper presents a compelling model for using negative learning mechanism and prompt learning, a discussion on whether NeGo can handle any of the complex environment shift challenges. For example, when the environment does not refer to the size of the graph but to other complex cases, whether the proposed method is still valid.

2. It is well known that prompt learning has achieved success in many fields. Therefore, I believe it is important to explore in more detail whether prompt learning or negative learning specifically contributes to the performance improvements presented in this paper.

3. It would be beneficial to highlight studies that are closely related to this paper and allocate more space for a detailed discussion of these works in Section 3.

**Questions For Authors:**

1. Whether the model is valid after the environment factors in the graph becomes more complex?

2. Which plays a more critical role, prompt learning or negative inference?

**Relation To Broader Scientific Literature:**

The contributions of this work are both technical and innovative, and it is an excellent scientific literature.

**Theoretical Claims:**

Yes, I have checked the correctness of all proofs for theoretical claims. All theoretical analysis is relatively solid.

---

> ### Author Rebuttal · Authors · 2025-03-31
>
> Dear Reviewer cMu6,
>
> We appreciate your time on reviewing our work. We will provide further clarification to address your concerns.
>
> **W1 and Q1. Complex environments.**
>
> Our proposed NeGo remains effective in other complex environment scenarios, not limited to the size-dependent environment cases. Actually, we have discussed the effectiveness of NeGo in scenarios where spurious subgraph styles become more complex in Sec 6.3. As shown in Tab. 3, we observe the superior performance of NeGo in style-dependent complex environment cases, achieving a 1.29% performance improvement.
>
>
> **W2 and Q2. Prompt learning and negative learning.**
>
> Prompt learning and negative learning collaboratively contribute to performance improvement, which has been studied in Apendix E.4. Given your valuable suggestion, we will move the discussion from the Appendix to the main text in the revised version of our manuscript. Specifically, we designed two variants of NeGo: PoGo uses a positive learning mechanism, and PoGo (w/o. prompt) masks the prompt learning module. The performance comparison results we obtained are as follows: NeGo > PoGo > PoGo (w/o. prompt). Therefore, using only negative learning mechanism or only the prompt module does not achieve the best performance.
>
>
> **W3. Detailed discussion of related works.**
>
> We will emphasize the researches on graph OOD from an environment perspective with more details in Sec 3.
>
> *GALA investigates the problem of invariant graph representation learning from the perspective of environment augmentation. LECI introduces the environment discrimination module and utilizes an adversarial learning mechanism to achieve invariant learning. Our proposed NeGo share the same research goal with those works, achieving better invariant learning mechanisms through environment modeling. However, our research persepctive and techniques for achieving this goal are completely different with them. Specifically, we not only focus on environment shift issues in linear mixed scenarios, but also place greater emphasis on model performance under complex and nonlinear environment shifts. Technically, we use prompt learning and negative learning mechanisms to capture the environment.*
>
> Thank you again for your positive comments on our work, which are invaluable for improving our manuscript. We look forward to your further feedback.

---

### Decision · Program_Chairs · 2025-05-01

**Decision:**

Accept (poster)

**Comment:**

In the initial submission, the reviewers maintained a generally positive attitude toward both the content and quality of this manuscript. Following the authors' detailed responses, the reviewers' overall sentiment remained highly favorable. I believe this manuscript makes a valuable contribution to the community. After careful evaluation and consideration, I am inclined to accept this paper.